# INCREMENTAL SEQUENCE LEARNING

**Edwin D. de Jong**
Department of Information and Computing Sciences
Utrecht University
`https://edwin-de-jong.github.io/`

## ABSTRACT

Deep learning research over the past years has shown that by increasing the scope or difficulty of the learning problem over time, increasingly complex learning problems can be addressed. We study incremental learning in the context of sequence learning, using generative RNNs in the form of multi-layer recurrent Mixture Density Networks. While the potential of incremental or curriculum learning to enhance learning is known, indiscriminate application of the principle does not necessarily lead to improvement, and it is essential therefore to know *which forms* of incremental or curriculum learning have a positive effect. This research contributes to that aim by comparing three instantiations of incremental or curriculum learning.

We introduce *Incremental Sequence Learning*, a simple incremental approach to sequence learning.

Incremental Sequence Learning starts out by using only the first few steps of each sequence as training data. Each time a performance criterion has been reached, the length of the parts of the sequences used for training is increased.

To evaluate Incremental Sequence Learning and comparison methods, we introduce and make available a novel sequence learning task and data set: predicting and classifying MNIST pen stroke sequences, where the familiar handwritten digit images have been transformed to pen stroke sequences representing the skeletons of the digits.

We find that Incremental Sequence Learning greatly speeds up sequence learning and reaches the best test performance level of regular sequence learning 20 times faster, reduces the test error by 74%, and in general performs more robustly; it displays lower variance and achieves sustained progress after all three comparison methods have stopped improving. The two other instantiations of curriculum learning do not result in any noticeable improvement. A trained sequence prediction model is also used in transfer learning to the task of sequence classification, where it is found that transfer learning realizes improved classification performance compared to methods that learn to classify from scratch.

## 1 INTRODUCTION

### 1.1 INCREMENTAL LEARNING, TRANSFER LEARNING, AND REPRESENTATION LEARNING

Deep learning research over the past years has shown that by increasing the scope or difficulty of the learning problem over time, increasingly complex learning problems can be addressed. This principle has been described as *Incremental learning* by Elman (1991), and has a long history. Schlimmer and Granger (1986) described a pseudo-connectionist distributed concept learning approach involving incremental learning. Elman (1991) defined Incremental Learning as an approach where the training data is not presented all at once, but incrementally; see also Elman (1993). Giraud-Carrier (2000) defines Incremental Learning as follows: "A learning task is incremental if the training examples used to solve it become available over time, usually one at a time." Bengio et al. (2009) introduced the framework of Curriculum Learning. The central idea behind this approach is that a learning system is guided by presenting gradually more and/or more complex concepts. A formal definition is provided specifying that the distribution over examples converges monotonically towards the target training

distribution, and that the entropy of the distributions visited over time, and hence the diversity of training examples, increases.

An extension of the notion of incremental learning is to also let the *learning task* vary over time. This approach, known as Transfer Learning or Inductive Transfer, was first described by Pratt (1993). Thrun (1996) reported improved generalization performance for *lifelong learning* and described *representation learning*, whereas Caruana (1997) considered a Multitask learning setup where tasks are learned in parallel while using a *shared* representation. In coevolutionary algorithms, the coevolution of representations with solutions that employ them, see e.g. Moriarty (1997); de Jong and Oates (2002), provides another approach to representation learning. Representation learning can be seen as a special form of transfer learning, where one goal is to learn adequate representations, and the other goal, addressed in parallel or sequentially, is to use these representations to address the learning problem.

Several of the recent successes of deep learning can be attributed to representation learning and incremental learning. Bengio et al. (2013) provide a review and insightful discussion of representation learning. Parisotto et al. (2015) report experiments with transfer learning across Atari 2600 arcade games where up to 5 million frames of training time in each game are saved. More recently, successful transfer of robot learning from the virtual to the real world was achieved using transfer learning, see Rusu et al. (2016). And at the annual ImageNet Large-Scale Visual Recognition Challenge (ILSVRC), the depth of networks has steadily increased over the years, so far leading up to a network of 152 layers for the winning entry in the ILSVRC 2015 classification task; see He et al. (2015).

## 1.2 Sequence Learning

We study incremental learning in the context of *sequence learning*. The aim in sequence learning is to predict, given a step of the sequence, what the next step will be. By iteratively feeding the predicted output back into the network as the next input, the network can be used to produce a complete sequences of variable length. For a discussion of variants of sequence learning problems, see Sun and Giles (2001); a more recent treatment covering recurrent neural networks as used here is provided by Lipton (2015).

An interesting challenge in sequence learning is that for most sequence learning problems of interest, the next step in a sequence does not follow unambiguously from the previous step. If this were the case, i.e. if the underlying process generating the sequences satisfies the Markov property, the learning problem would be reduced to learning a mapping from each step to the next. Instead, steps in the sequence may depend on some or all of the preceding steps in the sequence. Therefore, a main challenge faced by a sequence learning model is to capture relevant information from the part of the sequence seen so far. This ability to capture relevant information about future sequences it may receive must be *developed* during training; the network must learn the ability to build up internal representations which encode relevant aspects of the sequence that is received.

## 1.3 Incremental Sequence Learning

The dependency on the partial sequence received so far provides a special opportunity for incremental learning that is specific to sequence learning. Whereas the examples in a supervised learning problem bear no known relation to each other, the steps in a sequence have a very specific relation; later steps in the sequence can only be learned well once the network has learned to develop the appropriate internal state summarizing the part of the sequence seen so far. This observation leads to the idea that sequence learning may be expedited by learning to predict the first few steps in each sequence first and, once reasonable performance has been achieved and (hence) a suitable internal representation of the initial part of the sequences has been developed, gradually increasing the length of the partial sequences used for training.

A *prefix* of a sequence is a consecutive subsequence (a substring) of the sequence starting from the first element; e.g. the prefix $S_3$ of a sequence $S$ consists of the first 3 steps of $S$. We define *Incremental Sequence Learning* as an approach to sequence learning whereby learning starts out by using only a short prefix of each sequence for training, and where the length of the prefixes used for training is gradually increased, up to the point where the complete sequences are used. The structure of sequence learning problems suggests that adequate modeling of the preceding part of the sequence

is a requirement for learning later parts of the sequence; *Incremental Sequence Learning* draws the consequence of this by learning to predict the earlier parts of the sequences first.

## 1.4 RELATED WORK

In presenting the framework of Curriculum Learning, Bengio et al. (2009) provide an example within the domain of sequence learning, more specifically concerning language modeling. There, the vocabulary used for training on word sequences is gradually increased, i.e. the subset of sequences used for training is gradually increased; this is analogous to one of the comparison methods used here. Another specialization of Curriculum Learning to the context of sequence learning described by Bengio et al. (2015) addresses the discrepancy between *training*, where the true previous step is presented as input, and *inference*, where the previous output from the network is used as input; with *scheduled sampling*, the probability of using the network output as input is adapted to gradually increase over time. Zaremba and Sutskever (2014) apply curriculum learning in a sequence-to-sequence learning context where a neural network learns to predict the outcome of Python programs. The generation of programs forming the training data is parameterized by two factors that control the complexity of the programs: the number of digits of the numbers used in the programs and the degree of nesting. While a number of different instantiations of incremental or curriculum learning have been described in the context of sequence learning, no clear guidance is available on which forms are effective. The particular form explored here of learning to predict the earlier parts of sequences first is straightforward, it makes use of the particular structure of sequence learning problems, and it is easy to implement; yet it has received very limited attention so far.

## 2 MNIST HANDWRITTEN DIGITS AS PEN STROKE SEQUENCES

### 2.1 MOTIVATION FOR REPRESENTING DIGITS AS PEN STROKE SEQUENCES

The classification of MNIST digit images, see LeCun and Cortes (2010), is one example of a task on which the success of deep learning has been demonstrated convincingly; a test error rate of 0.23% was obtained by Ciresan et al. (2012) using Multi-column Deep Neural Networks. To obtain a sequence learning data set for evaluating Incremental Sequence Learning, we created a variant of the familiar MNIST handwritten digit data set provided by LeCun and Cortes (2010) where each digit image is transformed into a sequence of pen strokes that could have generated the digit.

One motivation for representing digits as strokes is the notion that when humans try to discern digits or letters that are difficult to read, it appears natural to trace the line so as to reconstruct what path the author's pen may have taken. Indeed, Hinton and Nair (2005) note that the idea that patterns can be recognized by figuring out how they were generated was already introduced in the 1950's, and describe a generative model for handwritten digits that uses two pairs of opposing springs whose stiffnesses are controlled by a motor program.

Pen stroke sequences also form a natural and efficient representation for digits; handwriting constitutes a canonical manifestation of the manifold hypothesis, according to which "real-world data presented in high dimensional spaces are expected to concentrate in the vicinity of a manifold $\mathcal{M}$ of much lower dimensionality $d_{\mathcal{M}}$, embedded in high dimensional input space $\mathbb{R}^{d_x}$"; see Bengio et al. (2013). Specifically: (i) the vast majority of the pixels are white, (ii) almost all digit images consist of a single connected set of pixels, and (iii) the shapes mostly consist of smooth curved lines. This suggests that collections of pen strokes form a natural representation for the purpose of recognizing digits.

The relevance of the manifold hypothesis can also be appreciated by considering the space of all 2-D 28x28 binary pixel images; when sampling uniformly from this space, one is likely to only encounter images resembling TV noise, and the chances of observing any of the 70000 MNIST digit images is astronomically small. By contrast, a randomly generated pen stroke sequence is not unlikely to resemble a part of a digit, such as a short straight or curved line segment. This increased alignment of the digit data with its representation in the form of pen stroke sequences implies that the amount of computation required to address the learning problem can potentially be vastly reduced.

## 2.2 CONSTRUCTION OF THE PEN STROKE SEQUENCE DATA SET

The MNIST handwritten digit data set consists of 60000 training images and 10000 test images, each forming 28 x 28 bit map images of written numerical digits from 0 to 9. The digits are transformed into one or more pen strokes, each consisting of a sequence of pen offset pairs $(dx, dy)$. To extract the pen stroke sequences, the following steps are performed:

1. Incremental thesholding. Starting from the original MNIST grayscale image, the following characteristics are measured:

   - The number of nonzero pixels
   - The number of connected components, for both the 4-connected and 8-connected variants.

   Starting from a thresholding level of zero, the thresholding level is increased stepwise, until either (A) the number of 4-connected or 8-connected components changes, (B) the number of remaining pixels drops below 50% of the original number, or (C) the thresholding level reaches a preselected maximum level (250). When any of these conditions occur, the previous level (i.e. the highest thresholding level for which none of these conditions occurred) is selected.

2. A common method for image thinning, described by Zhang and Suen (1984), is applied.

3. After the thresholding and thinning steps, the result is a skeleton of the original digit image that mostly consists of single-pixel-width lines.

4. Finding a pen stroke sequence that could have produced the digit skeleton can be viewed as a Traveling Salesman Problem where, starting from the origin, all points of the digit skeleton are visited. Each point is represented by the pen offset $(dx, dy)$ from the previous to the current point. For any transition to a non-neighboring pixel (based on 8-connected distance), an extra step is inserted with $(dx, dy) = (0, 0)$ and with eos = 1 (end-of-stroke), to indicate that the current stroke has ended and the pen is to be lifted off the paper. At the end of each sequence, a final step with values (0, 0, 1, 1) is appended. The fourth value represents eod, end-of-digit. This final tuple of the sequence marks that both the current stroke and the current sequence have ended, and forms a signal that the next input presented to the network will belong to another digit.

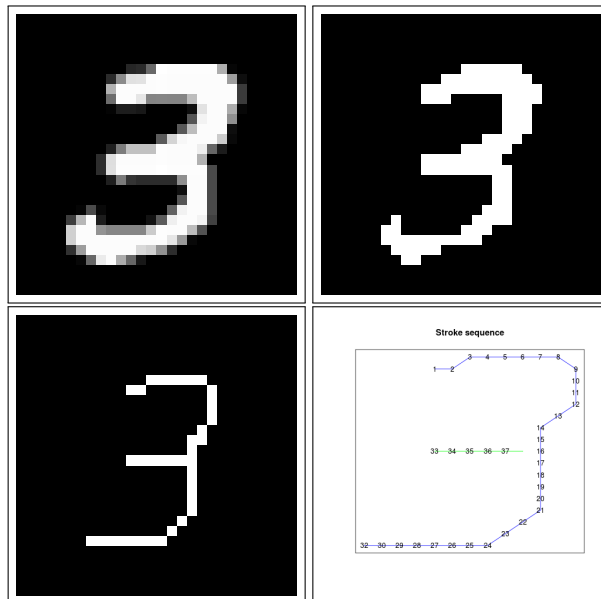

Figure 1: The original image (top left), thresholded image, thinned image, and actual extracted pen stroke image.

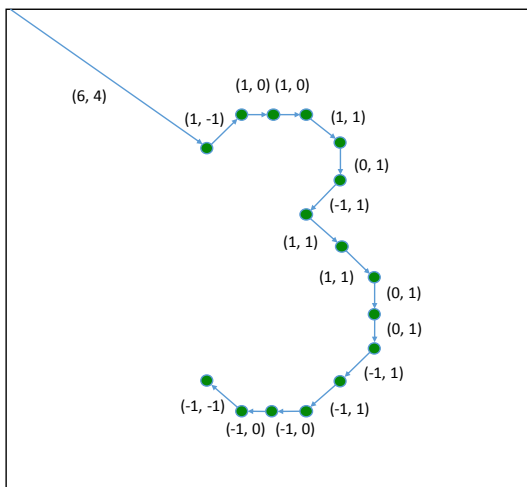

| $dx$ | $dy$ | eos | eod |
|---|---|---|---|
| 6 | 4 | 0 | 0 |
| 1 | -1 | 0 | 0 |
| 1 | 0 | 0 | 0 |
| 1 | 0 | 0 | 0 |
| 1 | 1 | 0 | 0 |
| 0 | 1 | 0 | 0 |
| -1 | 1 | 0 | 0 |
| 1 | 1 | 0 | 0 |
| 1 | 1 | 0 | 0 |
| 0 | 1 | 0 | 0 |
| 0 | 1 | 0 | 0 |
| -1 | 1 | 0 | 0 |
| -1 | 1 | 0 | 0 |
| -1 | 0 | 0 | 0 |
| -1 | 0 | 0 | 0 |
| -1 | -1 | 0 | 0 |
| 0 | 0 | 1 | 1 |

Figure 2: Example of a pen stroke image.

Table 1: Corresponding sequence. The origin is at the top left, and the positive vertical direction is downward. From the origin to the first point, the first offset is 6 steps to the right and 4 down: (6, 4). Then to the second point: 1 to the right and 1 up, (1, -1); etc.

It is important to note that the thinning operation discards pixels and therefore information; this implies that the sequence learning problem constructed here should be viewed as a new learning problem, i.e. performance on this new task cannot be directly compared to results on the original MNIST classification task. While for many images the thinned skeleton is an adequate representation that retains original shape, in other cases relevant information is lost as part of the thinning process.

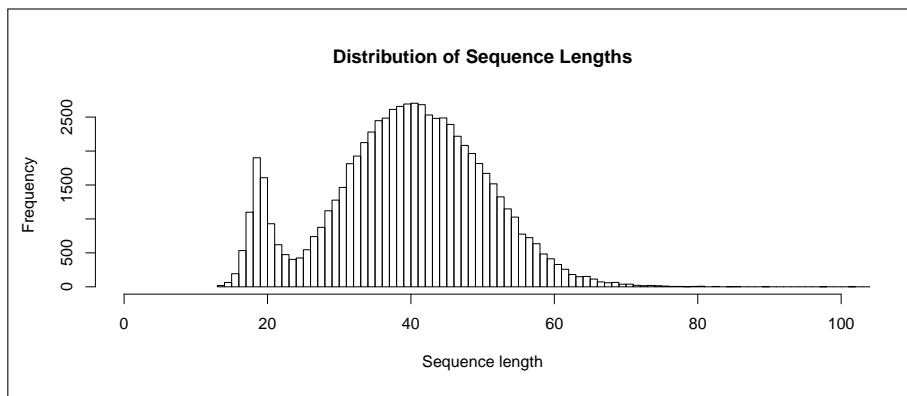

Figure 3: Distribution of sequence lengths. The average sequence length is approximately 40 steps.

## 3 NETWORK ARCHITECTURE

We adopt the approach to generative neural networks described by Graves (2013) which makes use of *mixture density networks*, introduced by Bishop (1994). One *sequence* corresponds to one complete image of a digit skeleton, represented as a sequence of $\langle dx, dy, eos, eod \rangle$ tuples, and may contain one or more strokes; see previous section.

The network has four input units, corresponding to these four input variables. To produce the input for the network, the $(dx, dy)$ pairs are scaled to yield two real-valued input variables $dx$ and $dy$. The

variables indicating the end-of-stroke (EOS) and end-of-digit (EOD) are binary inputs. Two hidden LSTM layers, see Hochreiter and Schmidhuber (1997), of 200 units each are used.

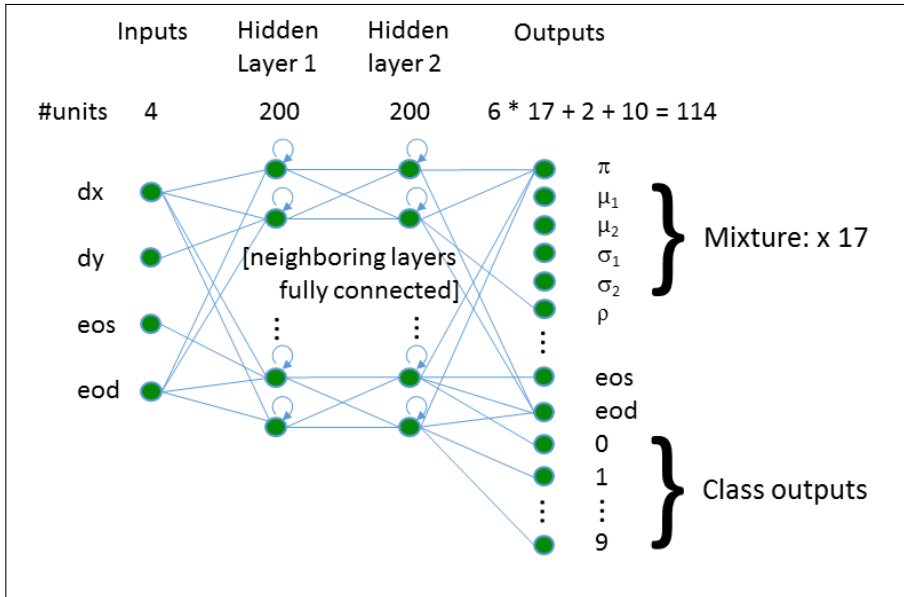

Figure 4: Network architecture; see text.

The input units receive one step of a sequence at a time, starting with the first step. The goal for the output units is to predict the immediate next step in the sequence, but rather than trying to directly predict $dx$ and $dy$, the output units represent a mixture of bivariate Gaussians. The output layer consists of the end of stroke signal (EOS), and a set of means $\mu^i$, standard deviations $\sigma^i$, correlations $\rho^i$, and mixture weights $\pi^i$ for each of the $\mathcal{M}$ mixture components, where the number of mixture components $\mathcal{M} = 17$ was found empirically to yield good results and is used in the experiments presented here. Additionally, a binary indicator signaling the end of digit (EOD) is used, to mark the end of each sequence. In addition to these output elements for predicting the pen stroke sequences, 10 binary class variable outputs are added, representing the 10 digit classes. This facilitates switching the task from sequence prediction to sequence classification, as will be discussed later; the output of these units is ignored in the sequence prediction experiments. The number of output units depends on the number of mixture components used, and equals $6\mathcal{M} + 2 + 10 = 114$.

For regularization, we found in early experiments that using the *maximum* weight as a regularization term produced better results than using the more common L-2 regularization. This approach can be viewed as L-∞-norm regularization, and has been used previously in the context of regularization, see e.g. Schmidt et al. (2008).

The definition of the sequence prediction loss $\mathcal{L}_P$ follows Graves (2013), with the difference that terms for the eod and for the L-∞ loss are included:

$$\mathcal{L}(\mathbf{x}) = \sum_{t=1}^{T} -\log\left(\sum_j \pi_t^j \mathcal{N}(x_{t+1}|\mu_t^j, \sigma_t^j, \rho_t^j)\right) \quad - \quad \begin{cases} \log eos_t & if\ (x_{t+1})_3 = 1 \\ \log(1 - eos_t) & otherwise \end{cases}$$

$$-\begin{cases} \log eod_t & if\ (x_{t+1})_4 = 1 \\ \log(1 - eod_t) & otherwise \end{cases} \quad + \quad \lambda||w||_\infty$$

## 4  INCREMENTAL SEQUENCE LEARNING AND COMPARISON METHODS

Below we describe Incremental Sequence Learning and three comparison methods, where two of the comparison methods are other instantiations of curriculum learning, and the third comparison is regular sequence learning without a curriculum learning aspect.

- Regular sequence learning
  The baseline method is regular sequence learning; here, all training data is used from the outset.

- Incremental Sequence Learning: increasing sequence length
  Predicting the second step of a sequence given the first step is a straightforward mapping problem that can be handled using regular supervised learning methods. The prediction of later steps in the sequence can potentially depend on all preceding steps, and for some cases may only be learned once an effective internal representation has been developed that summarizes relevant information present in the preceding part of the sequence. For predicting the $17^{th}$ step for example, the available input consist of the previous 16 steps, and the network must learn to construct a compact representation of the preceding steps that have been seen. More specifically, it must be able to distinguish between subspaces of the sequence space that correspond to different distributions for the next step in the sequence. The number of possible contexts grows exponentially with the position in the sequence, and the task of summarizing the preceding sequence therefore potentially becomes more difficult as a function of the position within the sequence. The problem of learning to predict steps later on in the sequence is therefore potentially much harder than learning to predict the earlier steps. In Incremental Sequence Learning therefore, the length of sequences presented to the network is increased as learning progresses.

- Increasing training set size
  Bengio et al. (2009) describe an application of curriculum learning to sequence learning, where the task is to predict the best word which can follow a given context of words in a correct English sentence. The curriculum strategy used there is to grow the vocabulary size. Transferring this to the context of pen stroke sequence generation, the most straightforward translation is to use subsets of the training data that grow in size, where the order of examples that are added to the training set is random.

- Increasing number of classes
  The network is first presented with sequences from only one digit class; e.g. all *zeros*. The number of classes is increased until all 10 digit classes are represented in the training data.

All three curriculum learning methods employ a threshold criterion based on the training RMSE; once a specified level of the RMSE has been reached, the set of training examples (determined by the number of sequence steps used, the number of sequences used, or the number of digits) is increased. We note that many possible variants of this simple adaptive scheme are possible, some of which may provide improvements of the results.

## 5    EXPERIMENTAL SETTINGS

In this section, we describe the experimental setup in detail.

The configuration of the baseline method, regular sequence learning, is as follows. The number of mixture components $\mathcal{M} = 17$, two hidden layers of size 200 are used. A batch size of 50 sequences per batch is used in these first experiments. The learning rate is $\alpha = 0.0025$, with a decay rate of 0.99995 per epoch. The order of training sequences (not steps within the sequences) is randomized. The weight of the regularization component $\lambda = 0.25$. In these first experiments, a subset of 10 000 training sequences and 5 000 test sequences is used. The error measure in these figures is the RMSE of the pen offsets (unscaled) predicted by the network given the previous pen movement.

The RMSE is calculated based on the difference between the predicted and actual $(dx, dy)$ pairs, scaled back to their original range of pixel units, so as to obtain an interpretable error; the *eos* and *eod* components of the error, which do form part of the loss, are not used in this error measure. For the method where the sequence length is varied, the number of individual points (input-target pairs) that must be processed per sequence varies over the course of a run. The number of sequences processed (or collections thereof such as batches or epochs) is therefore no longer an adequate measure of computational expense; performance is therefore reported as a function of the number of points processed.

Details per method:

- Incremental Sequence Learning
  The initial sequence length is 2, meaning that the first two points of each sequence are used, i.e. after feeding the first point as input, the second point is to be predicted. Once the training RMSE drops below the threshold value of 4, the length is doubled, up to the point where it reaches the maximum sequence length.

- Increasing training set size
  The initial training set size is 10. Each time the RMSE threshold of 4 is reached, this amount is doubled, up to the point where the complete set of training sequences is used.

- Increasing number of digit classes
  The initial number of classes is 1, meaning that only sequences representing the first digit (zero) are used. Each time the RMSE threshold of 4 is reached, this amount is doubled, up to the point where all 10 digit classes are used.

# 6 EXPERIMENTAL RESULTS

## 6.1 SEQUENCE PREDICTION: COMPARISON OF THE METHODS

Figures 5 shows a comparison of the results of the four methods. The baseline method (in red) does not use curriculum learning, and is presented with the entire training set from the start. Incremental Sequence Learning (in green) performs markedly better than all comparison methods. It reaches the best test performance of the baseline methods *twenty times faster*; see the horizontal dotted black line. Moreover, Incremental Sequence Learning greatly improves generalization; on this subset of the data, the average test performance over 10 runs reaches 1.5 for Incremental Sequence Learning vs 3.9 for regular sequence learning, representing a reduction of the error of 74%.

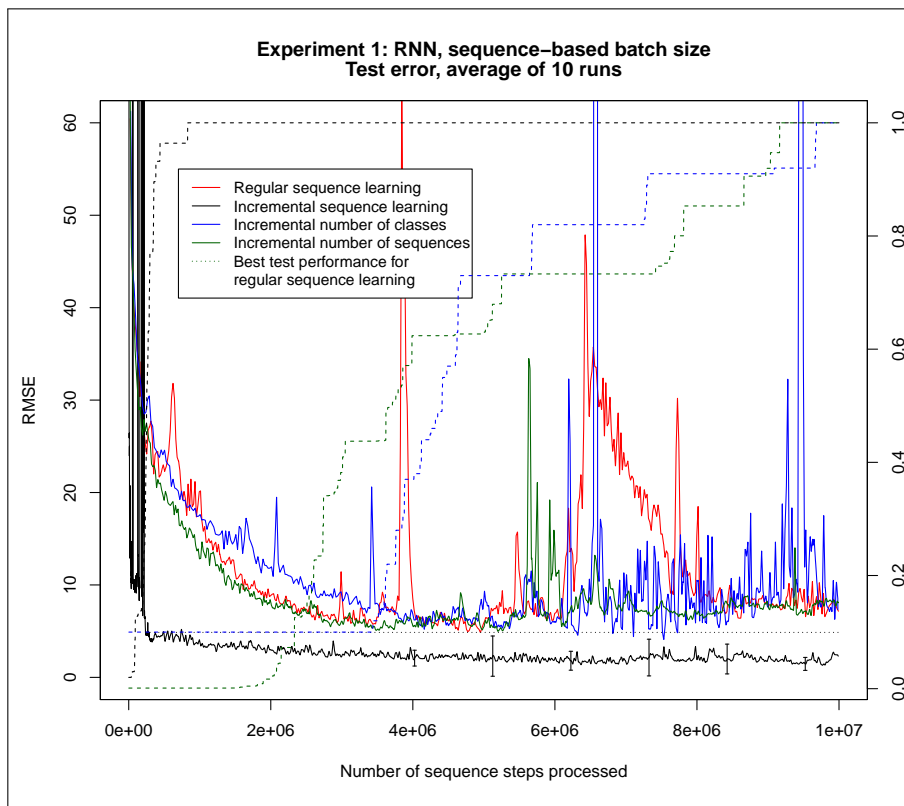

Figure 5: Comparison of the test error of the four methods, averaged over ten runs. The dotted lines indicate, at each point in time, which fraction of the training data has been made available at that point for the method of the corresponding color.

We furthermore note that the *variance* of the test error is substantially lower than for each of the other methods, as seen in the performance graphs; and where the three comparison methods reach their best test error just before $4 \cdot 10^6$ processed sequence steps and then begin to deteriorate, the test error for incremental sequence learning continues to steadily decrease over the course of the run.

| Method | Test set error |
|---|---|
| Regular sequence learning | 7.82 |
| Incremental sequence learning | 2.06 |
| Incremental number of classes | 7.64 |
| Incremental number of sequences | 6.27 |

Table 2: Best value for the average over 10 runs of the test set error obtained by each of the methods in Experiment 1. Incremental Sequence Learning achieves a reduction of 74% compared to regular sequence learning.

The two other curriculum methods do not provide any speedup or advantage compared to the baseline method, and in fact result in a *higher* test error; indiscriminate application of the curriculum learning principle apparently does not guarantee improved results and it is important therefore to discover *which forms* of curriculum learning can confer an advantage.

To explain the dramatic improvement achieved by Incremental Sequence Learning, we consider two possible hypotheses:
$\mathcal{H}1$: The number of sequences per batch is fixed (50), but the number of sequence *steps* or points varies, and is initially much smaller (2) for Incremental Sequence Learning. Thus, when measured in terms of the number of *points* that are being processed, the batch size for Incremental Sequence Learning is initially much smaller than for the remaining methods, and it increases adaptively over time. Hypothesis $\mathcal{H}1$ therefore is that (A) the smaller batch size improves performance, see Keskar et al. (2016) for earlier findings in this direction, and/or (B) the adaptive batch size aspect has a positive effect on performance.
$\mathcal{H}2$: Effectively learning later parts of the sequence requires an adequate internal representation of the preceding part of the sequence, which must be learned first; this formed the motivation for the Incremental Sequence Learning method.

To test the first hypothesis, $\mathcal{H}1$, we design a second experiment where the batch size is no longer defined in terms of the number of sequences, but in terms of the number of points or sequence steps, where the number of points is chosen such that the expected total number of points for the baseline method remains the same. Thus, whereas a batch for regular sequence learning contains 50 sequences of length 40 on average yielding 2000 points, Incremental Sequence Learning will start out with batches containing 1000 sequences of 2 points each, yielding the same total number of points.

Figure 6 shows the results. This change reduces the speedup during the earlier part of the runs, and thus partially explains the improvements observed with Incremental Sequence Learning. However, part of the speedup is still present, and moreover the three other observed improvements remain:

- Incremental Sequence Learning still features strongly improved generalization performance

- Incremental Sequence Learning still has a much lower variance of the test error

- Incremental Sequence Learning still continues improving at the point where the test performance of all other methods starts deteriorating

In summary, the adaptive and initially smaller batch size of Incremental Sequence Learning explains part of the observed improvements, but not all. We therefore test to what extent hypothesis $\mathcal{H}2$ plays a role. To see whether the ability to first learn a suitable representation based on the earlier parts of the sequences plays a role, we compare the situation where this effect is ruled out. A straightforward way to achieve this is to use Feed-Forward Neural Networks (FFNNs); whereas Recurrent Neural Networks (RNNs) are able to learn such a representation by learning to build up relevant internal state, FFNNs lack this ability. Therefore if any advantage of Incremental Sequence Learning is seen when using FFNNs, it cannot be due to hypothesis $\mathcal{H}2$. Conversely, if using FFNNs removes the advantage, the advantage must have be due to the difference between FFNNs and RNNs, which exactly corresponds to the ability to build up an informative internal representation, i.e. $\mathcal{H}2$. Since

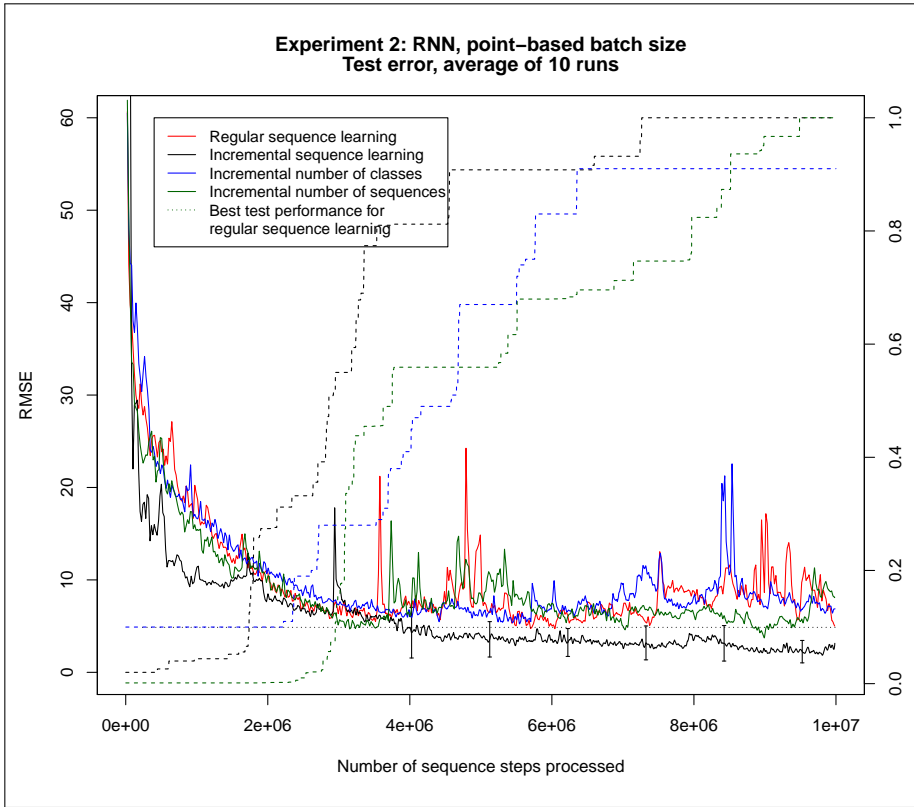

Figure 6: Comparison of the test error of the four methods, averaged over ten runs.

we want to explain the remaining part of the effect, we also use a batch size based on the number of points, as in Experiment 2.

Figure 7 shows the results. As the figure shows, when using FFNNs, the advantage of Incremental Sequence Learning is entirely lost. This provides a clear demonstration that both of the hypotheses $\mathcal{H}1$ and $\mathcal{H}2$ play a role. Together the two hypotheses explain the total effect of the difference, suggesting that the proposed hypotheses are also the only explanatory factors that play a role.

It is interesting to compare the performance of the RNN and their FFNN variants, by comparing the results of Experiments 2 and 3. From this comparison, it is seen that for Incremental Sequence Learning, the RNN variant achieves improved performance compared to the FFNN variant, as would be expected, since a FFNN cannot make use of any knowledge of the preceding part of the sequence and is thus limited to learning a general mapping between two subsequent pen offsets pairs $(dx_k, dy_k)$ and $(dx_{k+1}, dy_{k+1})$. However, it is the *only* method of the four to do so; for all three other methods, around the point where test performance for the RNN variants starts to deteriorate (after around $4 \cdot 10^6$ processed sequence steps), FFNN performance continues to improve and surpasses that of the RNN variants. This suggests that Incremental Sequence Learning is the only method that is able to utilize information about the preceding part of the sequence, and thereby surpass FFNN performance. In terms of absolute performance, a strong further improvement can be obtained by using the entire training set, as will be seen in the next section. These results suggest that learning the earlier parts of the sequence first can be instrumental in sequence learning.

## 6.2 Loss as a function of sequence position

To further analyze why variation of the sequence length has a particularly strong effect on sequence learning, we evaluate how the relative difficulty of learning a sequence step relates to the position within the sequence. To do so, we measure the average loss contribution of the points or steps within a sequence as a function of their position within the sequence, as obtained with a learning method that

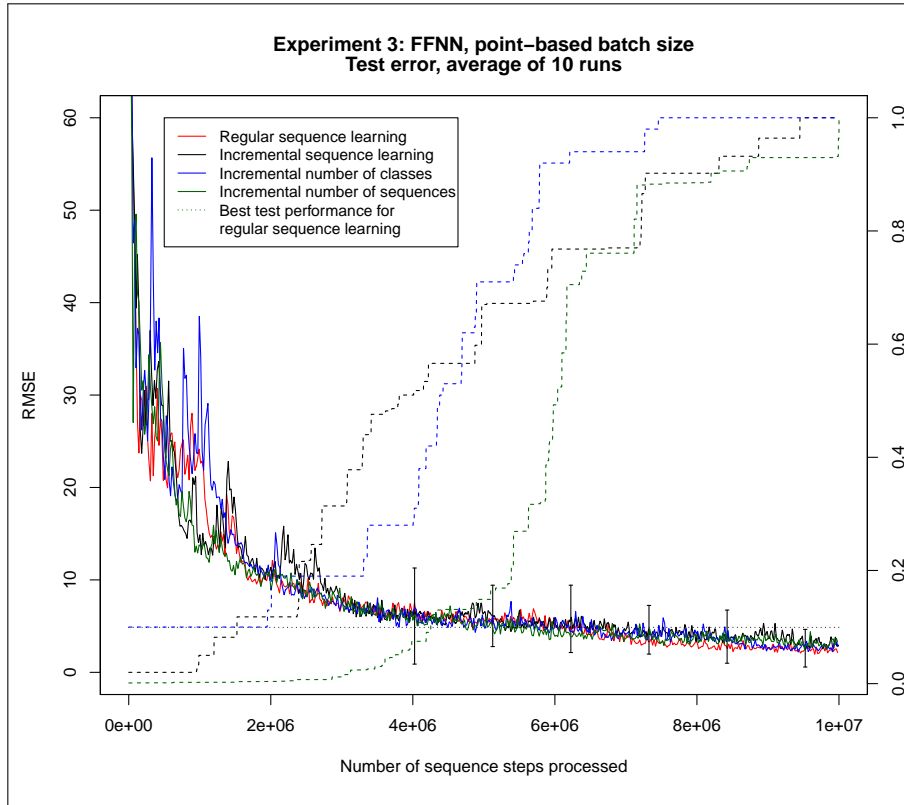

Figure 7: Comparison of the test error of the four methods, averaged over ten runs.

learns entire sequences (no incremental learning), averaged over the first hundred epochs of training. Figure 8 shows the results.

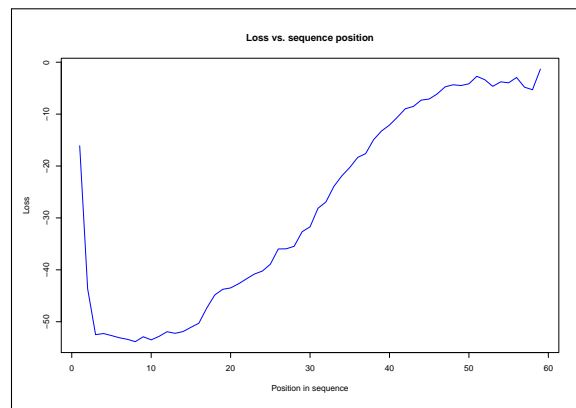

Figure 8: The figure shows the average loss contribution of the points or steps within a sequence as a function of their position within the sequence (see text). The first steps are fundamentally unpredictable. Once some context has been received, the loss for the next steps steeply drops. Later on in the sequence however, the loss increases strongly. This effect may be explained by the fact that the number of possible preceding contexts increases exponentially, thus posing stronger requirements on the learning system for steps later on in the sequence, and/or by the point that later parts of the sequences can only be learned adequately once earlier parts have been learned first, as later steps can depend on any of the earlier steps.

The first steps are fundamentally unpredictable as the network cannot know which example it will receive next; accordingly, at the start of the sequence, the error is high, as the method cannot know in advance what the shape or digit class of the new sequence will be. Once the first steps of the sequence have been received and the context increasingly narrows down the possibilities, the loss for the prediction of the next steps steeply drops. Subsequently however, as the position in the sequence advances, the loss increases strongly, and exceeds the initial uncertainty of the first steps. This effect may be explained by the fact that the number of possible preceding contexts increases exponentially, thus posing stronger requirements on the learning system for steps later on in the sequence.

## 6.3    RESULTS ON THE FULL MNIST PEN STROKE SEQUENCE DATA SET

The results reported so far were based on a subset of 10000 training sequences and 5000 test sequences, in order to complete a sufficient number of runs for each of the experiments within a reasonable amount of time. Given the positive results obtained with Incremental Sequence Learning, we now train this method on the full MNIST Pen Stroke Sequence Data Set, consisting of 60000 training sequences and 10000 test sequences (Experiment 4). In these experiments, a batch size of 500 sequences instead of 50 is used.

Figure 9 shows the results. Compared to the performance of the above experiments, a strong improvement is obtained by training on this larger set of examples; whereas the best test error in the results above was slightly above 1.5, the test performance for this experiment drops below one; a test error of 0.972 on the full test data set is obtained. An strking finding is that while initially the test error is much larger than the train error, the test error continues to improve for a long time, and approaches the training error very closely; in other words, no overtraining is observed even for relatively long runs where the training performance appears to be nearly converged.

## 6.4    TRANSFER LEARNING

The first task considered here was to perform sequence learning: predicting step t+1 of a sequence given step t. To adequately perform this task, the network must learn to detect which digit it is being fed; the initial part of a sequence representing a 2 or 3 for example is very similar, but as evidence is growing that the current sequence represents a 3, that information is vital in predicting how the stroke will continue.

Given that the network is expected to have built up some representation of what digit it is reading, an interesting test is to see whether it is able to switch to the task of sequence *classification*. The input presentation remains the same: at every time step, the recurrent neural network is fed one step of the sequence of pen movements representing the strokes of a digit. However, we now also read the output of the 10 binary class variable outputs. The target for these is a one-hot representation of the digit, i.e. the target value for the output corresponding to the digit is one, and all nine other target values are zero. To obtain the output, softmax is used, and the sequence classification loss $\mathcal{L}_C$ for the classification outputs is the cross entropy, weighted by a factor $\gamma = 10$:

$$\mathcal{L}_C = \gamma \left( -\frac{1}{N} \sum_{n=1}^{N} [y_n \log \hat{y}_n + (1 - y_n) \log(1 - \hat{y}_n)] \right)$$

In the following experiments, the loss consists of the sequence classification loss $\mathcal{L}_C$, to which optionally the earlier sequence prediction loss $\mathcal{L}_P$ is added, regulated by a binary parameter $\beta$:

$$\mathcal{L} = \mathcal{L}_C + \beta \mathcal{L}_P$$

The network is asked for a prediction of the digit class after each step it receives. Clearly, accurate classification is impossible during the first part of a sequence; before the first point is received, the sequence could represent any of the 10 digits with equal probability. As the sequence is received step by step however, the network receives more information. The prediction produced after receiving the one-but-last step of the sequence, i.e. at the point where the network was previously asked to predict the last step, is used as its final answer for predicting the digit class.

We compare the following variants:

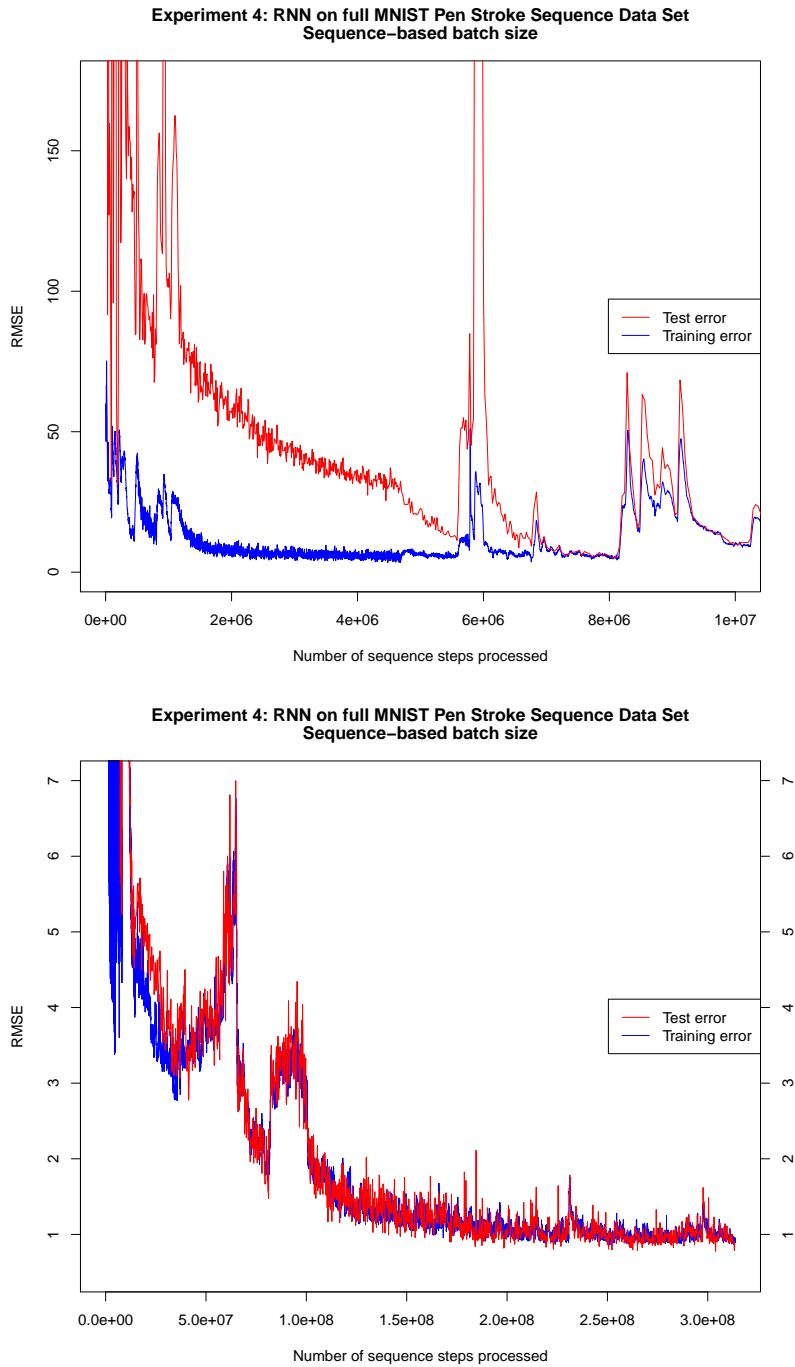

Figure 9: Performance on full MNIST Pen Stroke Sequence Data Set, zoomed to first part of the run and same experiment, results for the full run.

- Transfer learning: sequence classification and sequence prediction
  Starting from a trained sequence prediction model as obtained in Experiment 4, the earlier loss function is augmented with the sequence classification loss: $\mathcal{L} = \mathcal{L}_C + \mathcal{L}_P$

- Transfer Learning: sequence classification only
  Starting from a trained sequence prediction model, the loss function is switched such that it only reflects the classification performance, and no longer tracks the sequence prediction performance: $\mathcal{L} = \mathcal{L}_C$

- Learning from scratch, sequence classification and sequence prediction
  In this variant, learning starts from scratch, and both classification loss and prediction loss are used, as in the first experiment: $\mathcal{L} = \mathcal{L}_C + \mathcal{L}_P$

- Learning from scratch, sequence classification only
  $\mathcal{L} = \mathcal{L}_C$

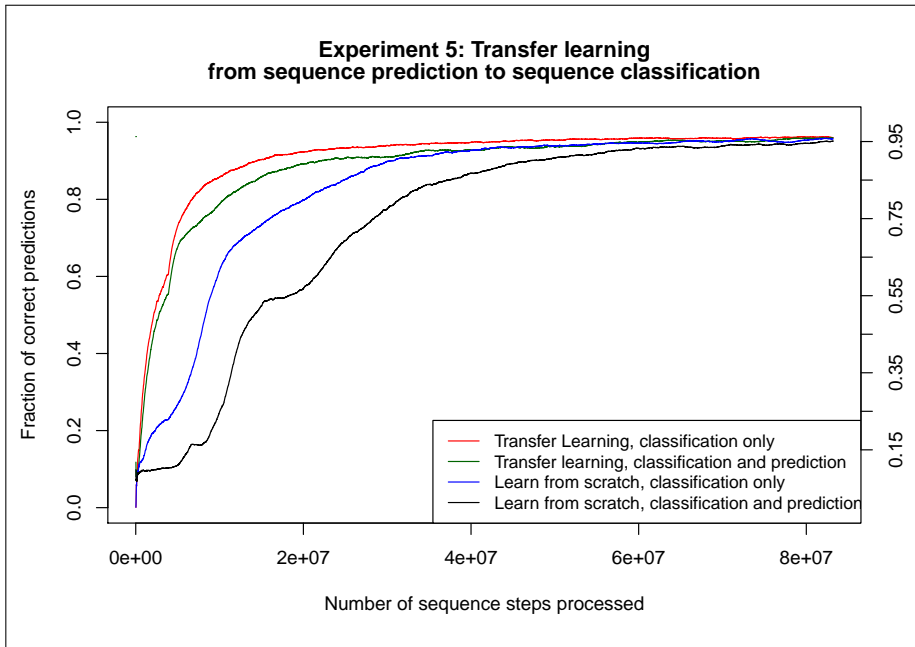

Figure 10: Using the sequence prediction model as a starting point for sequence classification: starting from a trained sequence prediction network, the task is switched to predicting the class of the digit (red and black lines). A comparison with learning a digit classification model from scratch (blue and green lines) shows that the internal state built up to predict sequence steps is helpful in predicting the class of the digit represented by the sequence.

Figure 10 shows the results; indeed the network is able to build further on its ability to predict pen stroke sequences, and learns the sequence classification task faster and more accurately than an identical network that learns the sequence classification task from scratch; in this first and straightforward transfer learning experiment based on the MNIST stroke sequence data set, a classification accuracy of 96.0% is reached[1]. We note that performance on the MNIST sequence data cannot be compared to results obtained with the original MNIST data set, as the information in the input data is vastly reduced. This result sets a first baseline for the MNIST stroke sequence data set; we expect there is ample room for improvement. Simultaneously learning sequence prediction and sequence classification does not appear to provide an advantage, neither for transfer learning nor for learning from scratch.

---

[1]This performance was reached after training for $7 \cdot 10^7$ sequence steps, i.e. roughly twice as long as the run shown in the chart

## 7 GENERATIVE RESULTS

To gain insight into what the network has learned, in this section we report examples of output of the network.

### 7.1 DEVELOPMENT DURING TRAINING

During training, the network receives each sequence step by step, and after each step, it outputs its expectation of the offset of the next point. In these figures and movies, we visualize the predictions of the network for a given sequence at different stages of the training process. All results have been obtained from a single run of Incremental Sequence Learning.

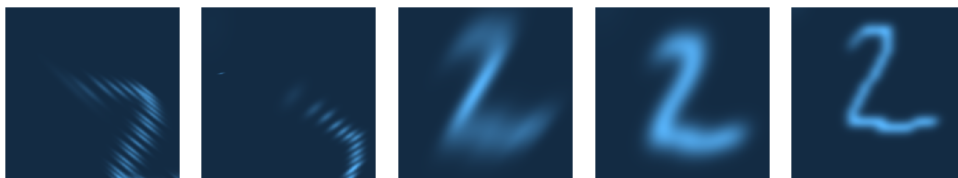

After 80 batches  After 140 batches  After 530 batches  After 570 batches  After 650 batches

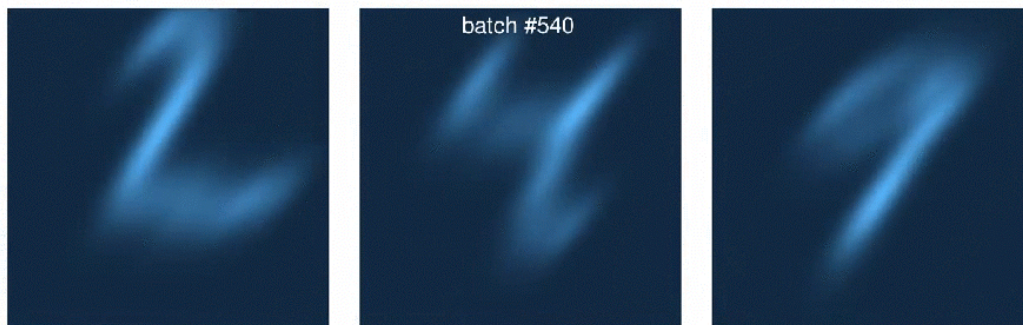

Figure 11: Movie showing what the network has learned over time. The movie shows the output for three sequences of the test data at different stages during training. To view, click the image or visit this link: https://edwin-de-jong.github.io/blog/isl/rnn-movies/generative-rnn-training-movie.gif.

### 7.2 UNGUIDED OUTPUT GENERATION, A.K.A. NEURAL NETWORK HALLUCINATION

After training, the trained network can be used to generate output independently. The guidance that is present during training in the form of receiving each next step of the sequence following a prediction is not available here. Instead, the output produced by the network is fed back into the network as its next input, see Figures 12 and 13. Figure 14 shows example results.

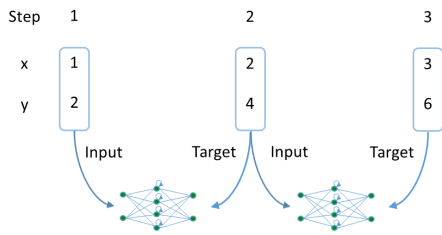 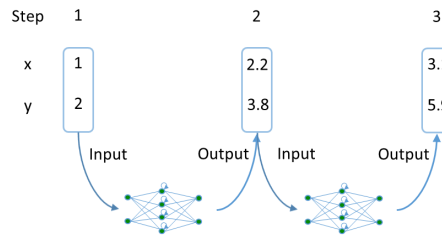

Figure 12: Training: the *target* of a training step is used as the next input.

Figure 13: Generation: the *output* of the network is used as the next input.

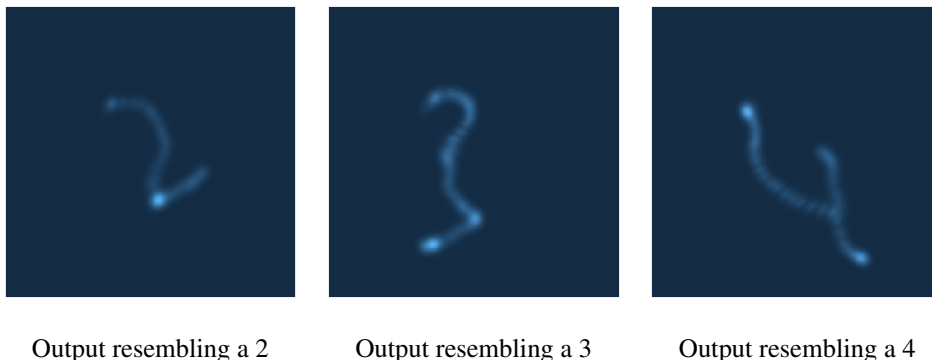

Output resembling a 2 Output resembling a 3 Output resembling a 4

Figure 14: Unguided output of the network: after each step, the network's output is fed back as the next input. Clearly, the network has learned the ability to independently produce long sequences representing different digits that occurred in training data.

## 7.3 SEQUENCE CLASSIFICATION

The third analysis of the behavior the trained network is to view what happens during sequence classification. At each step of the sequence, we monitor the ten class outputs and visualize their output. As more steps of the sequence are being received, the network receives more information, and adjusts its expectation of what digit class the sequence represents.

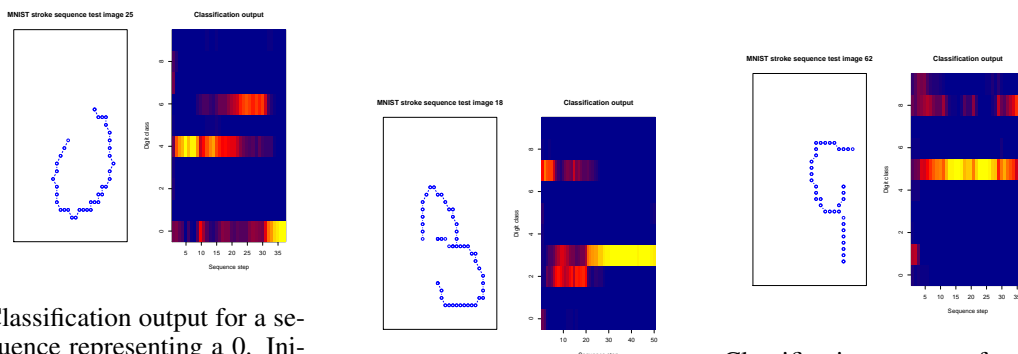

Classification output for a sequence representing a 0. Initially, as the downward part of the curved stroke is being received, the network believes the sequences represents a 4. After passing the lowest point of the figure, it assigns higher likelihood to a 6. Only at the very end, just in time before the sequence ends, the prediction of the network switches for the last time, and a high probability is assigned to the correct class.

Classification output for a sequence representing a 3. Initially, the networks estimates the sequence to represent a 7. Next, it expects a 2 is more likely. After 20 points have been received, it concludes (correctly) that the sequences represents a 3.

Classification output for a sequence representing a 9. While receiving the sequence, the dominant prediction of the network is that the sequence represents a five; the open loop of the 9 and the straight top line may contribute to this. When the last points are received, the network consider a 9 to be more likely, but some ambiguity remains.

## 8 CONCLUSIONS

There are many possible ways to apply the principles of incremental or curriculum learning to sequence learning, but so far a general understanding of *which forms* of curriculum sequence learning have a positive effect is missing. We have investigated a particular approach to sequence learning where the training data is initially limited to the first few steps of each sequence. Gradually, as the

network learns to predict the early parts of the sequences, the length of the part of the sequences used for training is increased. We name this approach Incremental Sequence Learning, and find that it strongly improves sequence learning performance. Two other forms of curriculum sequence learning used for comparison did not display improvements compared to regular sequence learning. The origins of this performance improvement are analyzed in comparison experiments, as detailed below.

A first observation was that with Incremental Sequence Learning, the time required to attain the best test performance level of regular sequence learning was much lower; on average, the method reached this level twenty times faster, thus achieving a significant speedup and reduction of the computational cost of sequence learning. More importantly, Incremental Sequence Learning was found to reduce the test error of regular sequence learning by 74%.

To analyze the cause of the observed speedup and performance improvements, we first increase the number of sequences per batch for Incremental Sequence Learning, so that all methods use the same number of sequence steps per batch. This reduced the speedup, but the improvement of the generalization performance was maintained. We then replaced the RNN layers with feed forward network layers, so that the networks can no longer maintain information about the earlier part of the sequences. This completely removed the remaining advantage. This provides clear evidence that the improvement in generalization performance is due to the specific ability of an RNN to build up internal representations of the sequences it receives, and that the ability to develop these representations is aided by training on the early parts of sequences first.

Next, we trained Incremental Sequence Learning on the full MNIST stroke sequence data set, and found that the use of this larger training set further improves sequence prediction performance. The trained model was then used as a starting point for transfer learning, where the task was switched from sequence *prediction* to sequence *classification*.

We conclude that Incremental Sequence Learning provides a simple and easily applicable approach to sequence learning that was found to produce large improvements in both computation time and generalization performance. The dependency of later steps in a sequence on the preceding steps is characteristic of virtually all sequence learning problems. We therefore expect that this approach can yield improvements for sequence learning applications in general, and recommend its usage, given that exclusively positive results were obtained with the approach so far.

## 9 RESOURCES

The Tensorflow implementation that was used to perform these experiments is available here: `https://github.com/edwin-de-jong/incremental-sequence-learning`

The MNIST stroke sequence data set is available for download here: `https://github.com/edwin-de-jong/mnist-digits-stroke-sequence-data/wiki/MNIST-digits-stroke-sequence-data`

The code for transforming the MNIST digit data set to a pen stroke sequence data set has also been made available: `https://github.com/edwin-de-jong/mnist-digits-as-stroke-sequences/wiki/MNIST-digits-as-stroke-sequences-(code)`

ACKNOWLEDGMENTS

The author would like to thank Max Welling, Dick de Ridder and Michiel de Jong for valuable comments and suggestions on earlier versions.

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
