# Peer review of "Incremental Sequence Learning"

_ICLR 2017 — rejected_

[Public Comment · Tara N Sainath · 07 Nov 2016]
**ICLR Paper Format**

Dear Authors,

Please resubmit your paper in the ICLR 2017 format for your submission to be considered. Thank you!

[Author Response · Edwin D. de Jong · 09 Nov 2016]
**Updated version**

Updated version:
- added a section with generative results, including movies showing what the network has learned over the course of training
- used the ICLR style file

[Author Response · Edwin D. de Jong · 01 Dec 2016]
**Updated version December 1 (please use this version)**

- Clarified the contribution (see abstract, intro, and conclusion)
- Added figures to illustrate the architecture of the network and the difference between training and generation
- Adapted the selection of experiments in Section 6.4 (more logical selection of experimental settings)
- Some textual edits

Article length: the current version of the article aims to provide a description of the work that is as clear and complete as possible. As a result the number of pages is higher than requested for final versions (this is in line with the ICLR submission policy, which permits submitting longer articles). If accepted, a condensed version will be produced that satisfies the ICLR recommended guidelines ("8 pages, plus 1 page for the references and as many pages as needed in an appendix section").

[Reviewer Comment · AnonReviewer1 · 08 Dec 2016]
**Max of AVG**

You mention the "Best value for the average over 10 runs".
Can you explain what you calculated here?

[Official Review · AnonReviewer1 · rating 5 · confidence 4 · 16 Dec 2016]
**Interesting idea, long experiments, but only a single non-standard dataset**

The submitted paper proposes a new way of learning sequence predictors. In the lines of incremental learning and curriculum learning, easier samples are presented first and the complexity is increased during training. The particularity here is that the complexity is defined as the length of the sequences given for training, the premise being is that longer sequences are harder to learn, since they need a more complex internal representation.

The targeted application is sequence prediction from primed prefixes, tested on a single dataset, which the authors extract themselves from MNIST.

The idea in the paper is interesting and worth reading. There are also many interesting aspects of evaluation part, as the authors perform several ablation studies to rule out side-effects of the tests. The proposed learning strategy is compared to other strategies.

However, my biggest concern is still with evaluation. The authors tested the method on a single dataset, which is non standard and derived from MNIST. Given the general nature of the claim, in order to confirm the interest of the proposed algorithm, it need to be tested on other datasets, public datasets, and on a different application.

The paper is too long and should be trimmed significantly.

The transfer learning part (from prediction to classification) is a different story and I do not see a clear connection to the main contribution of the paper.

The presentation and organization of the paper could be improved. It is quite sequentially written and sometimes reads like a student's report.

The loss given in the long unnumbered equation on page 6 should be better explained: provide explanations for each term, and make clearer what the different symbols mean. Learning is supervised, so which variables are predictions, and which are observations from the data (ground truth).

Names in table 2 do not correspond to the descriptions in section 4.

[Official Review · AnonReviewer4 · rating 5 · confidence 3 · 16 Dec 2016 (modified: 21 Jan 2017)]
**No Title**

This paper presents a thorough analysis of different methods to do curriculum learning. The major issue I have with it is that the dataset used seems very specific and does not necessarily justified, as mentioned by AnonReviewer3. It would have been great to see experiments on more standard tasks. Also, I really can't understand how the performance of FFNN models can be so good, please elaborate on this (see last comment).
However, the paper is well written, the comparisons of the described methods are interesting and would probably apply to some other datasets as well.

The paper is way too long (18 pages!). Please reduce it or move some of the results to an appendix section.

The method described is extremely similar to the one described in Reinforcement learning neural turing machines (Zaremba et al., 2016,

[Official Review · AnonReviewer3 · rating 3 · confidence 4 · 20 Dec 2016]
**Really long paper with not a lot of impact**

First up, I want to point out that this paper is really long. Like 17 pages long -- without any supplementary material. While ICLR does not have an official page limit, it would be nice if authors put themselves in the reviewer's shoes and did not take undue advantage of this rule. Having 1 or 2 pages in addition to the conventional 8 page limit is ok, but more than doubling the pages is quite unfair. 

Now for the review: The paper proposes a new artificial dataset for sequence learning. I call it artificial because it was artificially generated from the original MNIST dataset which is a smallish dataset of real images of handwritten digits. In addition to the dataset, the authors propose to train recurrent networks using a schedule over the length of the sequence, which they call "incremental learning". The experiments show that their proposed schedule is better than not having any schedule on this data set. Furthermore, they also show that their proposed schedule is better than a few other intuitive schedules. The authors verify this by doing some ablation studies over the model on the proposed dataset. 

I have following issues with this paper: 

-- I did not find anything novel in this paper. The proposed incremental learning schedule is nothing new and is a natural thing to try when learning sequences. Similar idea have already been tried by a number of authors, including Bengio 2015, and Ranzato 2015. The only new piece of work is the ablation studies which the authors conduct to tease out and verify that indeed the improvement in performance is due to the curriculum used. 

-- Furthermore, the authors only test their hypothesis on a single dataset which they propose and is artificially generated. Why not use it on a real sequential dataset, such as, language modeling. Does the technique not work in that scenario? In fact I am quite positive that for language modeling where the vocabulary size is huge, the performance gains will be no where close to the 74% reported in the paper.

-- I'm not convinced about the value of having this artificial dataset. Already there are so many real world sequential dataset available, including in text, speech, finance and other areas. What exactly does this dataset bring to the table is not super clear to me. While having another dataset may not be a bad thing in itself, I almost felt that this dataset was created for the sole purpose of making the proposed ideas work. It would have been so much better had the authors shown experiments on other datasets. 

-- As I said, the paper is way too long. A significant part of the length of the paper is due to a collection of experiments which are completely un-related to the main message of the paper. For instance, the experiment in Section 6.2 is completely unrelated to the story of the paper. Same is true with the transfer learning experiments of Section 6.4.

[Author Response · Edwin D. de Jong · 28 Jan 2017]
**Scope of application**

Two reviewers had questions about which problems the approach is expected to work for. I did not find the time to do further experiments, but I did in the mean time think about when the approach can be expected to confer benefit. When there are long range dependencies where it is necessary to learn relations with a substantial part of the history, it is to be expected that gradually growing the length of the history over which relations are learned can bring benefit. 
As noted, I am willing to reduce the paper to the suggested 8 pages, and use an appendix for remaining material.

[Final Decision · Program Chairs · 06 Feb 2017]
**ICLR committee final decision**

This is an empirical paper which compares three different instantiations of a kind of incremental/curriculum learning for sequences.
 
 The reviews from R1 and R3 (which gave confidence scores of 4) were negative. The main concerns addressed by the reviewers:
 * Paper is too long -- 17 pages -- and length is due to experiments (e.g. transfer learning) which are tangential to the main message of the paper (R3, R1) 
 * Lack of novelty (R3)
 * Tests only on single, synthetic, small dataset and questioning the claim that this new synthetic dataset is helpful to the community (R3, R1)
 
 However, R3 and R1 both pointed out that they found the ablation studies interesting. R4 (who gave a confidence score of 3) was gave a more positive score but also expressed similar concerns with R1 & R3 (page length, similarity to existing work, dataset too specific and not necessarily justified).
 
 The author argued for the novelty of the paper, agreed to reduce the paper length and also argued that the data was indeed helpful (giving a specific case of another researcher who was extending the data). The author also provided a "twitter trail" countering the argument that the dataset was created for the sole purpose of showing that the method works.
 
 After engaging the reviewers in discussion, R4 admitted they were originally too generous with their score and downgraded to 5. The AC has decided that, while the paper has merits as acknowledged by the reviewers, it's not strong enough for acceptance in its present form. The AC encourages the author to work on an improved version (perhaps with experiments on an additional real dataset) and organize it with the audience in mind.